# Time-Stability Dispersion of MWCNTs for the Improvement of Mechanical Properties of Portland Cement Specimens

**DOI:** 10.3390/ma13184149

**Published:** 2020-09-18

**Authors:** Laura M. Echeverry-Cardona, Natalia Álzate, Elisabeth Restrepo-Parra, Rogelio Ospina, Jorge H. Quintero-Orozco

**Affiliations:** 1Laboratorio de Física del Plasma, Universidad Nacional de Colombia, Sede Manizales, Manizales 170001, Colombia; lmecheverryc@unal.edu.co (L.M.E.-C.); nalzatea@unal.edu.co (N.Á.); 2Laboratory of Biological Materials Science and Semiconductors, Universidad Industrial de Santander, Bucaramanga 681012, Colombia; rospinao@uis.edu.co (R.O.); jhquinteroo@gmail.com (J.H.Q.-O.)

**Keywords:** MWCNTs, dispersion, sonication energy, stability, cement

## Abstract

This study shows the energy optimization and stabilization in the time of solutions composed of H_2_O + TX-100 + Multi-Wall Carbon Nanotubes (MWCNTs), used to improve the mechanical properties of Portland cement pastes. For developing this research, sonication energies at 90, 190, 290, 340, 390, 440, 490 and 590 J/g are applied to a colloidal substance (MWCNTs/TX-100 + H_2_O) with a molarity of 10 mM. Raman spectroscopy analyses showed that, for energies greater than 440 J/g, there are ruptures and fragmentation of the MWCNTs; meanwhile at energies below 390 J/g, better dispersions are obtained. The stability of the dispersion over time was evaluated over 13 weeks using UV-vis spectroscopy and Zeta Potential. With the most relevant data collected, sonication energies of 190, 390 and 490 J/g, at 10 mM were selected at the first and the fourth week of storage to obtain Portland cement specimens. Finally, we found an improvement of the mechanical properties of the samples built with Portland cement and solutions stored for one and four weeks; it can be concluded that the MWCNTs improved the hydration period.

## 1. Introduction

One of the primary materials in architectural and infrastructure projects is concrete; for this reason, it is necessary to search for new options for improving and optimizing its physical, chemical and mechanical properties. The inclusion of nano additives [1], nanostructured materials [2,3,4], nano reinforcements [5] and hybridization of hydrates with organic molecules [6] improves the mechanical features of the cement matrix through the interaction with calcium silicate hydrate (C-S-H) cementitious [7]. Meanwhile, Multi-Wall Carbon Nanotubes (MWCNTs) have demonstrated a significant improvement of mechanical resistance in the cementitious matrix, due to the filling of microcracks, pores and bonds between the surface and the matrix [8,9,10,11,12]. Besides, it enables the transduction of a mechanical strain into a change of the electrical resistance and it transformed into a self-sensing cement-based material, which would have great potential for structural health monitoring, conditions evaluation for concrete structures, highway traffic monitoring, border security and structural vibration control, among others. However, for improving their conductance, it is essential to know how to properly disperse MWCNT within the cement matrix, since the highly attractive Van der Waals forces make the MWCNTs agglomerated [13,14,15,16,17,18].

Solutions, as the implementation of chemical and mechanical dispersion alternatives, guarantee dispersion stability over time [19,20,21]. Among the actual technologies, the application of ultrasound—where acoustic cavitation generates high shear stress—fragmenting the agglomeration of MWCNTs has been used [22]. Besides, the chemical approaches used can be covalent or non-covalent treatments which alter the surface energy of the wall of the MWCNTs, making possible their dispersion [23]. The combination of these methods has been widely used in recent years, obtaining more productive and stable dispersions [24,25,26]. On the other hand, we have studied three types of surfactants with different tensoactives (sodium dodecyl sulfate (anionic), cetylpyridinium chloride (ionic) and triton TX-100 (no ionic)) to improve the dispersion of MWCNTs. It was reported that Triton TX-100 was the most effective in lowering the surface tension of water, reaching its maximum effect at the 10 mM concentration [27,28,29]. Although the benefits of combining these methods are known, there are not enough studies on the time that dispersion remains stable before being used in the cement paste, which is essential to avoid the agglomeration of MWCNTs [24,25].

In this study, we show a follow-up on the MWCNT dispersions obtained by applying ultrasonic waves and the addition of a non-ionic surfactant. We observe their behavior over time through physical-chemical characterization techniques; furthermore, we evaluate the specimens of Portland cement through quasi-static compression tests.

## 2. Materials and Methods

Firstly, we made a dispersion of MWCNTs in water MilliQ (ultrapure water of laboratory) and Triton TX-100 surfactant through an ultrasonic tip and after that, we study the stabilization over time, taking spectra in a UV-Vis spectrophotometer (Shimadzu, Bucaramanga, Colombia) and Zetasizer Nano equipment (Malvern, Bucaramanga, Colombia), for 1, 2, 4, 10 and 13 weeks. The experiment was finished the thirteen weeks, because the MWCNTs were falling out abruptly in the solution. To evaluate the structural damage induced in the first week, Raman microscopy was used. Later, it was prepared two series of cement paste cylinders, at first and fourth weeks after the sonication process, observing that the mechanical properties are maintained. It was considered the fourth week since, for later weeks, the dispersion fall-out abruptly and the nanotubes were agglomerated. Finally, the elastic modulus of the cylinders was obtained through HM 5030 Master Loader equipment (Humboldt, Manizales, Colombia). Experimental parameters of the dispersion were based on some previous work carried out by us.

### 2.1. Dispersion of MWCNTs

The aqueous solution was composed of (i) 0.35 MWCNTs (Industrial grade NC7000 produced by Nanocyl, Sambreville, Belgium) [30], (ii) water MilliQ (ultrapure water for laboratory) and (iii) a concentration of 10 mM of Triton TX-100 (M = 646.85 g/mol, density = 1.060 ± 0.01 g/cm^3^) [31]. The surfactant was mixed with water MilliQ [32] by magnetic stirring for 5 min, at room temperature. This surfactant was chosen because, in previous work, we have found that its non-ionic activity is more favorable for this type of dispersion.

Then, the MWCNTs were added to the solution. The mixture was taken to the cabin of the ultrasonic tip to carry out the sonication. The energies applied by the ultrasonic tip were 90, 190, 290, 340, 390, 440, 490 and 590 J/g to optimize energy for ensuring an adequate degree of dispersion. The sonication process was carried out on a 500-watt ultrasonic processor (Sonics & Materials, Bucaramanga, Colombia) with 40% amplitude in 20 s on/off cycles to avoid overheating of the samples [33]. On the other hand, the sonication time was calculated following the work published by Mendoza-Reales [30]. The data are shown in Table 1.

Finally, the solutions were stored in 40 mL screw-cap bottles, at a temperature of 22 °C, in a dark place for taking spectra every 15 days.

### 2.2. Preparation of Cement Paste Cylinders

Cement paste cylinders were built with water-to-cement (w/c) ratio of 0.4 [33]. The MWCNTs and surfactant are part of the water ratio. The cement used was Portland Cement type 1. In previous work, we reported that an energy of 390 J/g guarantees an adequate dispersion degree; then, in this work, we chose energies around this value [32]. Besides, using the Raman analysis, we obtained three characteristic zones, which are represented by 190, 390 and 490 J/g dispersion energies and these energies were used to make the cement paste cylinders.

Cement paste cylinders were made in two families; the first was made after MWCNTs dispersion. Meanwhile, the second was made after storing the dispersion for four weeks. Before being used the dispersion, a bath is done in an ultrasonic cell for 600 s. In each family, twelve cement paste cylinders were made—(i) three for cement paste + H_2_O + TritonTX-100, (ii) three for cement paste with a solution of water + TX-100 + MWCNTs submitted to 190 J/g, (iii) three for cement paste with a solution of H_2_O + TX-100 + MWCNTs submitted to 390 J/g and (iv) three for cement paste with a solution of H_2_O + TX-100 + MWCNTs submitted to 490 J/g.

The cylinders were built following the NTC 550 (ASTM C305) and ASTM C109/C109 M standards with 1 inch in diameter [34,35]. For the filling process, three layers are placed on the mold, using the compaction method, tamping each layer 50 times and the vibration method with the help of a rubber mallet, giving 50 gentle blows to the edges of the mold. Finally, samples are dried in the open air at a temperature of 22 °C. The samples were removed from the mold after 24 h and cured in a lime saturated bath, as suggested in ASTM C192 [36].

### 2.3. Characterization Methods

A confocal Raman microscopy—LabRAM HR Evolution (Horiba, Bucaramanga, Colombia) was used to evaluate the structural damage induced on the MWCNTs with the following conditions—532 nm laser, optical microscope with 10× magnification and in a region of 1250–1690 cm^−1^. On the other hand, the evaluation of the dispersion stability was carried out using a Shimadzu UV-Vis UV2600 spectrophotometer, (Bucaramanga, Colombia) with a spectral range from 200 to 850 nm, equipped with a double beam and an integrating sphere. The spectra were measured in the 250 to 600 nm region and a Zetasizer Nano Series kit, Malvern Zetasizer nano ZEN 3690 reference (Bucaramanga, Colombia). The water was taken as a dispersant with a refractive index of 1.33 at 25 °C.

Finally, a Humbolt HM 5030 Master Loader (Manizales, Colombia) with a 50 kN capacity load cell was used in the test of specimens. The established parameters were a speed of 0.25 mm/min, taking the strain data every 0.010 mm and, with the load measured in kN for each strain reading [37]. The elasticity modulus and maximum strength were obtained from the stress-strain curve. Furthermore, to observe the distribution and anchorage of the MWCNTs with the cementitious matrix, a Carl Zeiss EVO MA 10 microscope (Medellin, Colombia) equipped with a 10-mm Oxford X-act silicon drift detector was used—the measurements were made with a working distance of 8.5 mm and energy of 10,000 and 20,000 kV.

## 3. Results

### 3.1. Analysis and Discussion

#### 3.1.1. Assessment of Induced Structural Damage on MWCNTs

Raman analysis showed the damage caused by the MWCNTs due to the sonication process for each energy used (see Figure 1). Three characteristic bands were observed—(i) the D band is due to the defect-induced phonon mode associated to the kinematic restriction disorders [38]. It means that this band contributed with a disorder, due to the presence of vacancies, defects or finite size of the MWCNTs structure, consisting of sp^2^ and sp^3^ bonds; (ii) G band is assigned to the phonon mode and it shows the vibration of the carbon-carbon sp^2^ bonds of the graphite structure present in the MWCNTs [39]; (iii) the G’ band is related to second-order dispersion processes; here, two phonons can be involved in the same way (overtone) or phonons in different modes (combination). This band has an origin similar to the D band [40]. Figure 2 shows a scheme of the vibration’s mode of the MWCNTs. The vibration mode E_2g_ band (Figure 2a) associated to G band, meanwhile, the vibration mode A1g (Figure 2b) associated to D band.

For developing the structural order analysis, an evaluation of the narrowing of the G band and the *I_G_*/*I_D_* ratio is taken into account, considering that the intensity of the D band decreases as the density of the defects decreases. Therefore, the *I_G_*/*I_D_* ratio was calculated since the deconvolution process concerning the higher intensity of the band and Lorentzian probability functions for the three characteristic bands (D, G, G’). With the *I_G_*/*I_D_* ratio, a relationship of the induced disorder with the increase in sonication energy was found. In Figure 3, the *I_G_*/*I_D_* ratio shows a decreasing trend when the sonication energy increases and it is associated with the ordered C-C bonds, which causes an inversely proportional effect with the *I_G_*/*I_D_* ratio. Three zones are observed—(i) the first zone between 90 and 290 J/g, shows a significant decrease in the *I_G_*/*I_D_* ratio from 1 to 0.75 [34]; this can be associated to the sonication process, which causes alterations in the structure of MWCNTs, because there are unstable molecules (as can be seen in Figure 4); (ii) In the second zone, from 290 to 440 J/g, the induced energy is greater than the energy between bonds π, which causes that the bonds break, without reaching the maximum energy allowed and (iii) in the third zone between 440 and 590 J/g, the *I_G_*/*I_D_* ratio decreases again, inducing a better dispersion. However, in the literature, these dispersions cannot be considered as the most optimal energies, because that can contribute to the breaking and fragmentation of MWCNTs. The induced energy affects the π and the sigma bonds of the structure. It implying loss of characteristic bonds that provide the unique properties of nanotubes [41].

#### 3.1.2. Assessment of Dispersion Stability

Through the UV-Vis absorbance spectrum changes in the degree of dispersion of each sample are observed, such that the agglomerated MWCNTs show absorption in the ultraviolet region. At the same time, the individual MWCNTs are active in the UV-Vis region; therefore, the degree of disentanglement and dispersion in the solution is related to the absorbance intensity [22,42]. Furthermore, the optical properties of MWCNTs depend on their one-dimensional (1D) character such as the confinement and quantization of the electronic and vibrational states of energy, depending on the radial direction, which gives rise to Van Hove singularities (vHss). These singularities are unique for each MWCNTs due to the variation of the diameter [43,44].

Figure 5 shows the changes of the absorbance spectrum taken the first week. The characteristic peak of the individual MWCNTs is observed at a wavelength of approximately 300 nm [45]. At sonication energies below 190 J/g, the peak intensity remains constant, while at energies above this value, the peak exhibits a significant increase. That phenomenon can be due to two simultaneous effects in the sonication process—(i) the deagglomeration of the MWCNTs and (ii) fragmentation of the individual MWCNTs [22].

Since it is not possible to keep the sonication constant throughout the process due to the fluctuations of voltage, temperature and mechanical energy delivered in the dispersion, it is necessary to trace absorbance vs. total energy curves. Figure 6 shows the evolution of the absorbance spectrum taken for each sample during week 1, 2, 4, 10 and 13. The absorbance intensity remains almost constant for each sonication energy. The sites of adsorption of the surfactant appear with higher interactions of Van der Walls and π–π, when the MWCNTs agglomerates are “unravelled” by a high local cut [46]. These interactions cause an increase in the π plasmon formation, which could imply an increase in the absorbance for weeks 1, 2 and 4. In the fourth week, an increase in absorbance values is observed, due to that the interaction of van der walls force is better between the MWCNTs than between the surface MWCNTs and the surfactant; this behavior would entail agglomerations of MWCNTs (verified in the zeta potential analysis).

Figure 7 shows the behavior determined using the zeta potential of the MWCNT/TX-100/H_2_O samples, varying the dispersion energy with constant molarity of 10 mM, over four weeks of storage. An increase in the zeta potential as the sonication energy increases is expected, since the energy increase leads to a more significant appearance of electrostatic charge on the surface of the nanotube [47]. However, for the energy of 290 J/g and 490 J/g, changes in the trend are observed. The evolution of 490 J/g is expected since the fragmentation of the nanotubes produces a more significant number of ends, providing additional electrostatic charges than to those present on the nanotube surface, which results in a greater repulsive force.

Another aspect that could have influenced the irregular behavior of samples is the variation of temperature (observed for cycles on/off in the dispersion process), which affects the zeta potential [48]. Additionally, a secondary effect is observed since, as the dispersion resting time in a non-ionic medium (TX-100) elapses, the absolute value of the zeta potential decreases and consequently, the stabilization of the particles dispersion also decreases, reaching the unstable electrical potential reported in the literature, +25 mV and −25 mV. This effect is related to the principle of minimum energy, as mentioned above. In the fourth week, the solution is found in an unstable zeta potential range; therefore, it was not continued with the analysis in the posterior weeks.

#### 3.1.3. Tests on Cement Specimens

Table 2 shows the results calculated from the quasi-static compression tests in specimens prepared with an MWCNTs/TX-100/H_2_O solution of one week of rest with variable sonication energies, at 190, 390 and 490 J/g. The calculated results are—(i) modulus of elasticity and (ii) maximum stress of the specimens during the curing process (7, 14 and 28 days). Table 2 also shows two increments in the properties, one associated with an increase in cure time and the other associated to an increase due to sonication energy [49,50].

Comparison of the effects produced on elasticity modulus of MWCNTs dispersed at different energies, during the curing time, with one week of preparation is shown in Table 2. It is evident that the addition of MWCNTs generates an increase in the elasticity modulus as follows—7.11±5.6% (190 J/g), 31.07±5.86% (390 J/g) and 66.05±17.36% (490 J/g). Meanwhile for maximum strength was of—49.44±16.73% (190 J/g), 57.08±12.71% (390 J/g) and 76.45±17.22% (490 J/g).

Results of the elasticity modulus and maximum strength of the specimens, including the dispersion at four weeks of rest and dispersed with energies of 190, 390 and 490 J/g during the respective curing process, are presented in Table 3. These data show a tendency to increase as dispersion energy and cure time increase.

Table 3 presents the effects on the elasticity modulus from the addition of MWCNTs in the test samples, at different dispersion energies (190, 390 and 490 J/g) and a time of rest of the four weeks. Differences between the elasticity modulus of the specimens with and without MWCNTs are 5.61±2.44% (190 J/g), 32.12±4.15% (390 J/g) and 65.07±15.92% (490 J/g). Meanwhile, for the maximum strength, changes are 29.72±6.28 (190 J/g), 44.97± 0.67 (390 J/g) and 81.59±20.71 (490 J/g).

In all the tests mentioned above, values of the mechanical properties (elasticity modulus and maximum strength) increase as a function of the energy, with a tendency of the 2nd-degree polynomial trendline, which is consistent with the following hypotheses:At energies of 490 J/g (at the first and the fourth week), the mechanical properties are stable, compared to other energies. For Raman analysis, in this energy range (≤440 J/g), the MWCNTs are affected in their structure (rupture of sigma bonds). An essential characteristic of composite materials for the improvement of their mechanical properties is the interaction between the phases of the compound. In the case of MWCNTs, there is a functional interaction because, when they are unravelled, there are radicals available to establish strong bonds within the compound [33].The increase in mechanical properties is also due to the low presence of micelle formation in the MWCNTs and, therefore, there is a lack of secondary bonds and Van der Waals forces which prevent the formation of energetically weak bonds. Besides, when is applying more energy to a solution, more carboxylic residues are formed. The transformation of sp2 bonds into sp3 hybridized carbon atoms [51], allows the interaction within the cementitious matrix with the C-S-H phase, generating a bridge effect, improving the load distribution and increasing the energy between bonds [52,53]. Additionally, the behavior of the maximum strength value may have some variability, due to the appearance of pores and defects during the synthesis of the materials and to systematic errors in the mechanical characterization process.The addition of MWCNTs accelerated the hydration during the curing process. They act as crystallization centers of cement hydrates and fill the voids between grains, resulting in the immobilization of free water molecules (phenomenon observed into scanning electron microscope (SEM) images, presented below) [54]. These nanocomposites allow generating a more compact C-S-H phase, in both grain surface and the porous space; this causes early hydration of the specimens and notably favors the mechanical properties during the curing time. The longer the curing time, the higher the hydration and, therefore, there will be better mechanical properties because the interaction phase between the compounds becomes stronger [55,56,57].

#### 3.1.4. Evaluation of the Morphology of Portland Cement Specimens with the Addition of MWCNTs

Images of the specimens prepared from the solution, with a constant molarity of surfactant and variable energies of 190, 390 and 490 J/g, were taken with a scanning electron microscope at resolutions of 2 and 10 µm.

The fracture surface of the samples (190, 390 and 490 J/g) was observed using SEM images, as presented in Figure 8, Figure 9 and Figure 10, in which, three ways of an arrangement of the MWCNTs are determined. The first is the agglomeration of MWCNTs around the hydrated pores; the second is the bridge effect, which occurs between one MWCNTs anchored to two cement fragments at their ends; and the last one occurs when the MWCNTs are scattered around the pores of the samples.

Figure 8 is taken from a test specimen prepared with a dispersed solution at 190 J/g. In this figure, it is shown a low presence of MWCNTs, cement fragments without anchoring and a grouping zone of MWCNTs that resembles a cobweb. It can be correlated with results obtained from the compression test and the physical-chemical analyzes. It is possible to say that the MWCNTs did not have a good dispersion in the TX-100 + H_2_O solution. Because at the time of using it as a reinforcement, it would influence the mechanical and microstructural performance of the cementitious matrices, without contributing to the reduction of porosity and the increase in stiffness [58].

On the other hand, in Figure 9, it is possible to note that there is a higher amount of dispersed MWCNTs. The 390 J/g dispersion energy causes MWCNT’s to deagglomerate in a higher proportion; it produces a reduction in the cement fragments without anchorage, dense areas of MWCNTs that resemble a cobweb, where the bridge effect is observed. Furthermore, a large number of MWCNTs with sufficient length to join capillary pores and act as crack bridges as a result of covalent bonds of MWCNTs. This process inhibits crack propagation, resulting in a better load capacity; moreover, ductility and fracture energy of specimens can be observed [59].

Figure 10 (490 J/g) shows the presence of shorter fragments of MWCNTs and fewer agglomerates, bridge effects and MWCNTs embedded in the hydrated matrix; in addition, no cuts are formed in a large area of cement. At higher energy, nanotubes disperse much more and break up, causing them to mix more homogeneously in the cement specimen, contrary to results shown in Figure 8, where dense points of MWCNTs that resemble a cobweb are observed and a more significant cement fractures are produced. Despite this, the energy of 490 J/g is not adequate because it influences the nanostructure of MWCNTs, which contributes to pore filling but at the same time, it produces a decrease in crack propagation, due to the MWCNTs length affects the mechanical performance. Additionally, these results agree with those obtained using physical-chemical techniques, in which at 490 J/g, there is a rupture of π and sigma bonds in the structure of MWCNTs.

## 4. Conclusions

MWCNTs exhibited good stability during the fourth week and a balance between the degree of dispersion desired; moreover, the damage induced was reached for 390 and 440 J/g dispersion energies. Meanwhile, for 90 and 290 J/g dispersion energies, there was not find a significant alteration in the interactions of the MWCNTs and the TX-100 + H_2_O solution.

Raman characterization showed structural damage and fragmentation of MWCNTs, which led to the presence of shorter fragments and lower agglomeration; therefore, this does not contribute to decrease of crack propagation length. The improvement of the mechanical properties of the cement paste with MWCNTs is due to binding energies generated between them, porosity reduction and the increase in the specimen stiffness.

Quasi-static compression tests (maximum strength and elasticity modulus) showed that the addition of MWCNTs has a positive effect on the mechanical properties of Portland cement specimens. When the cement pastes were built with a solution stored for four weeks, the mechanical properties (throughout the hydration period) remain almost constant, compared to the cement paste made during the first week.

Finally, for applications in civil engineering, it was found that the solution composed by H_2_O + TX-100-MWCNTs showed suitable stability when it was stored for four weeks, making this kind of research of potential interest for this industry.

## Figures and Tables

**Figure 1 materials-13-04149-f001:**
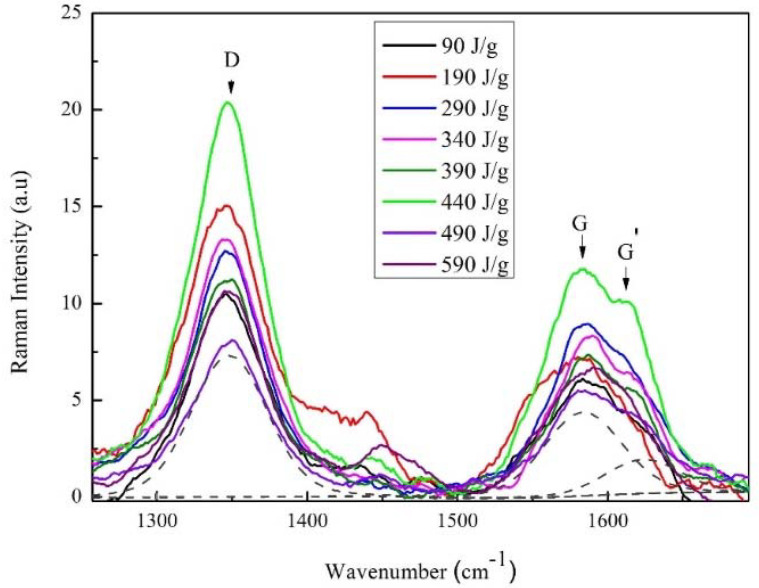
Raman spectra of Multi-Wall Carbon Nanotubes (MWCNTs)/TX-100/H_2_O at different energies of sonication; and the dotted lines show the deconvolution of the D, G and G’ bands.

**Figure 2 materials-13-04149-f002:**
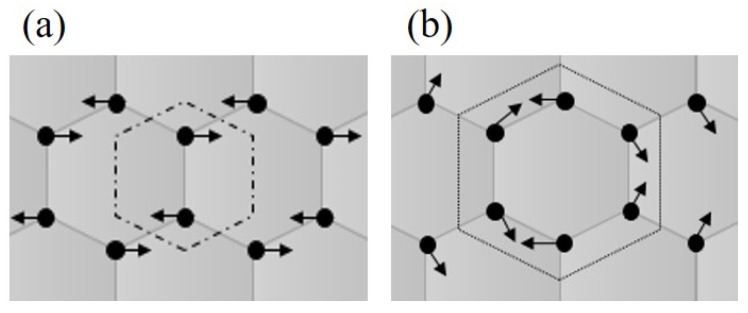
Vibration patterns with Mulliken notation. (**a**) Mode E2g. (**b**) Mode A1g.

**Figure 3 materials-13-04149-f003:**
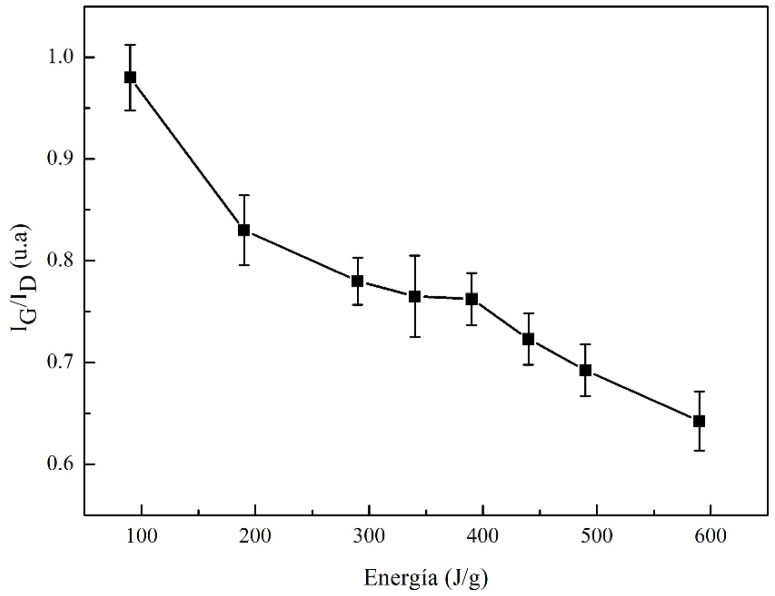
*I_G_/I_D_* relation of MWCNTs/TX-100/H_2_O at different sonicator energies.

**Figure 4 materials-13-04149-f004:**
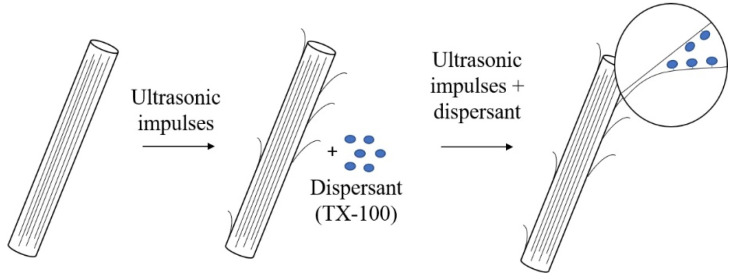
Gradual exfoliation process of MWCNT agglomerated scheme.

**Figure 5 materials-13-04149-f005:**
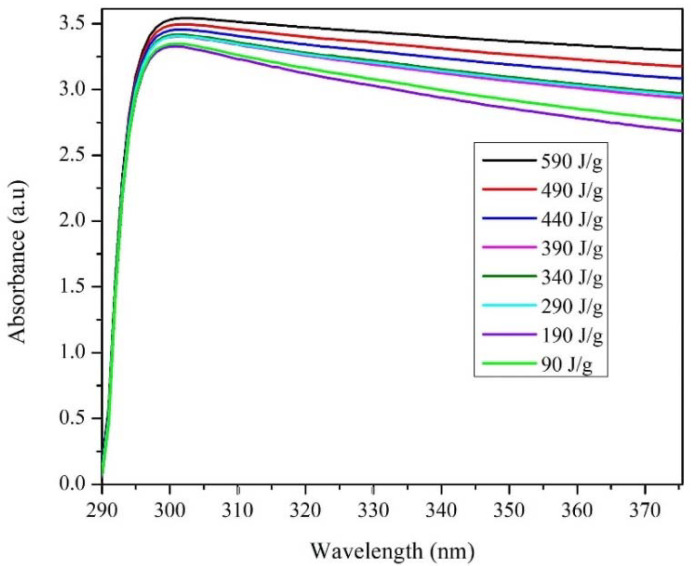
Evolution of UV-vis spectra in an aqueous solution at 0.35% in MWCNTs weight with 10 mM of TX-100 concentration as a function of sonicator energy.

**Figure 6 materials-13-04149-f006:**
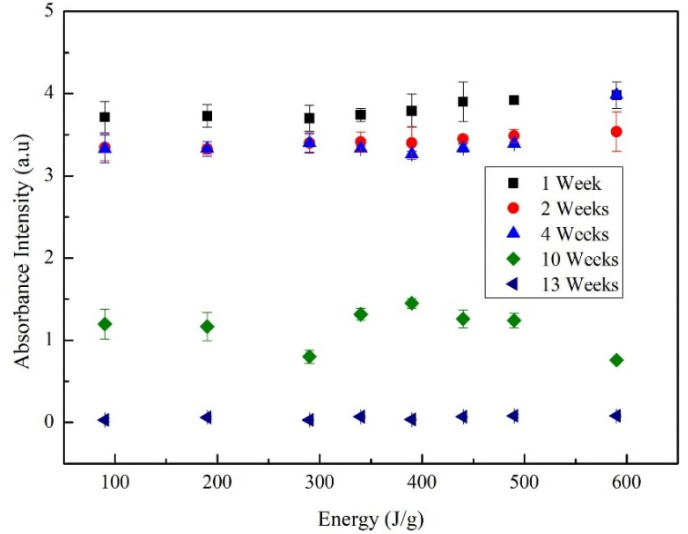
Evolution of absorbance at 300 nm of MWCNT/TX-100/H_2_O at different sonicator energies with constant molarity of 10 mM.

**Figure 7 materials-13-04149-f007:**
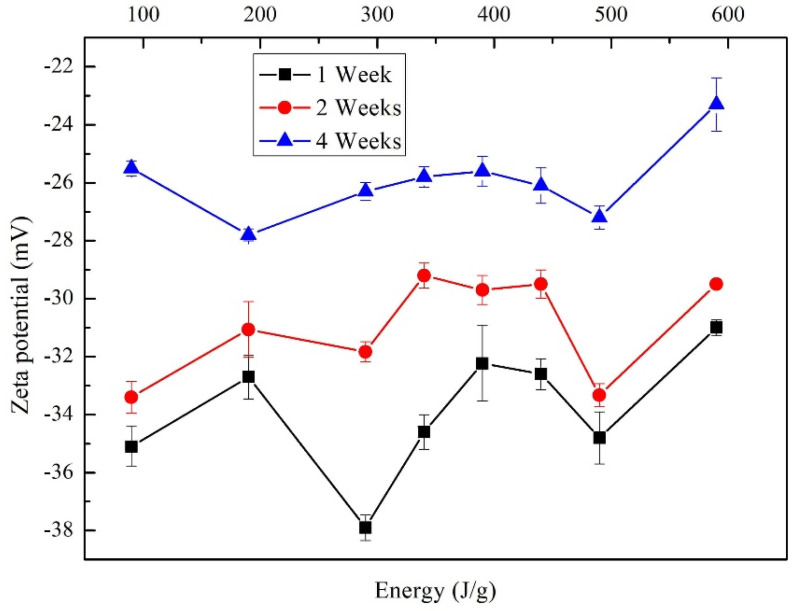
Zeta potential for samples MWCNTs/TX-100/H_2_O changing the dispersion energy.

**Figure 8 materials-13-04149-f008:**
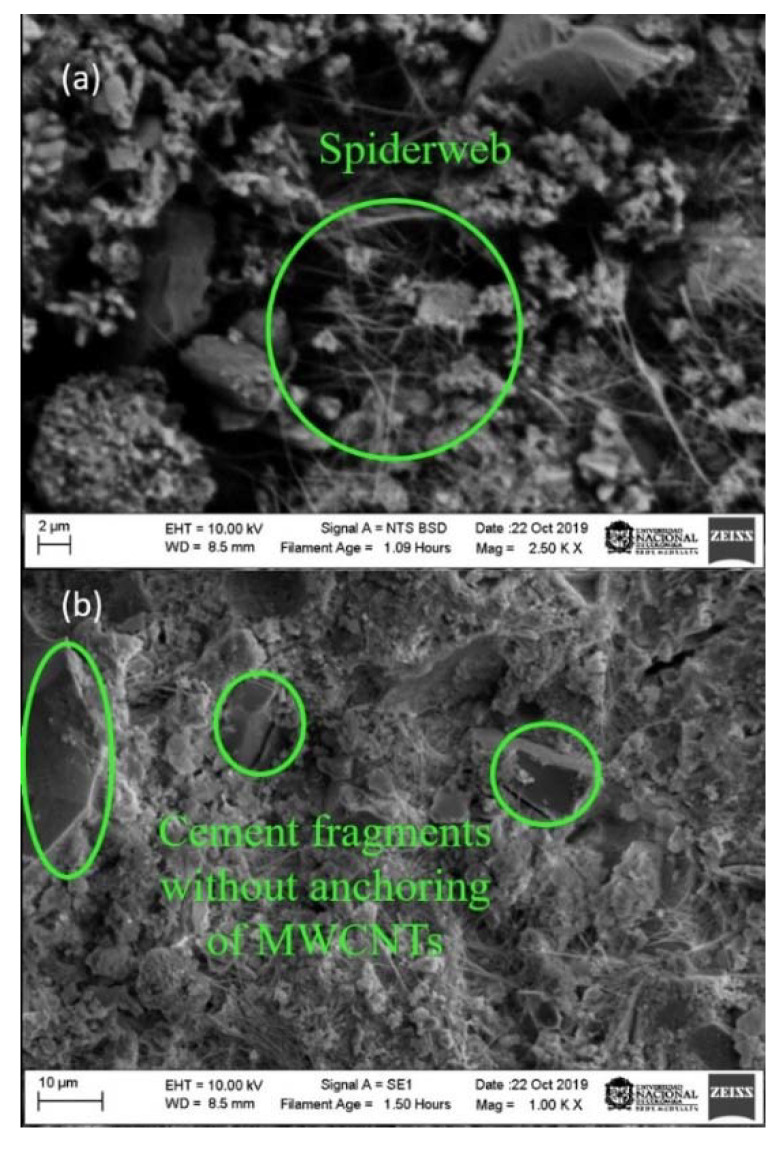
Scanning electron microscope (SEM) images of Portland cement specimens with a dispersion solution addition at 190 J/g. (**a**) 2 µm (**b**) 10µm.

**Figure 9 materials-13-04149-f009:**
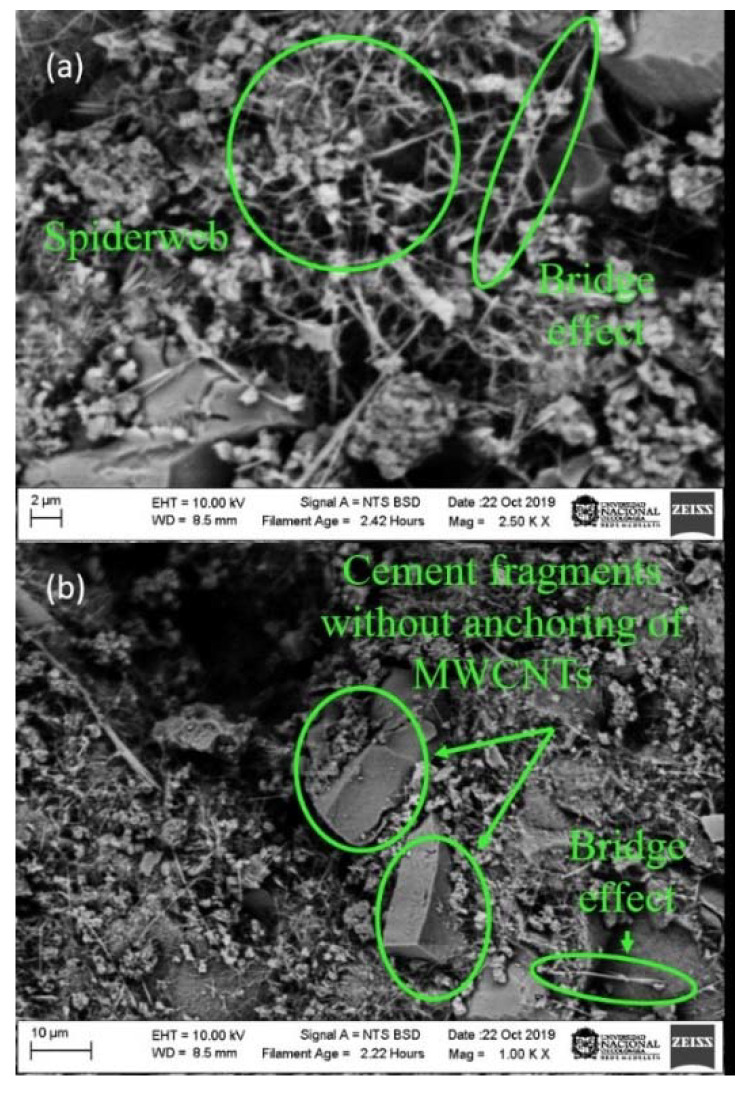
SEM images of Portland cement specimens with a dispersion solution addition at 390 J/g. (**a**) 2 µm (**b**) 10µm.

**Figure 10 materials-13-04149-f010:**
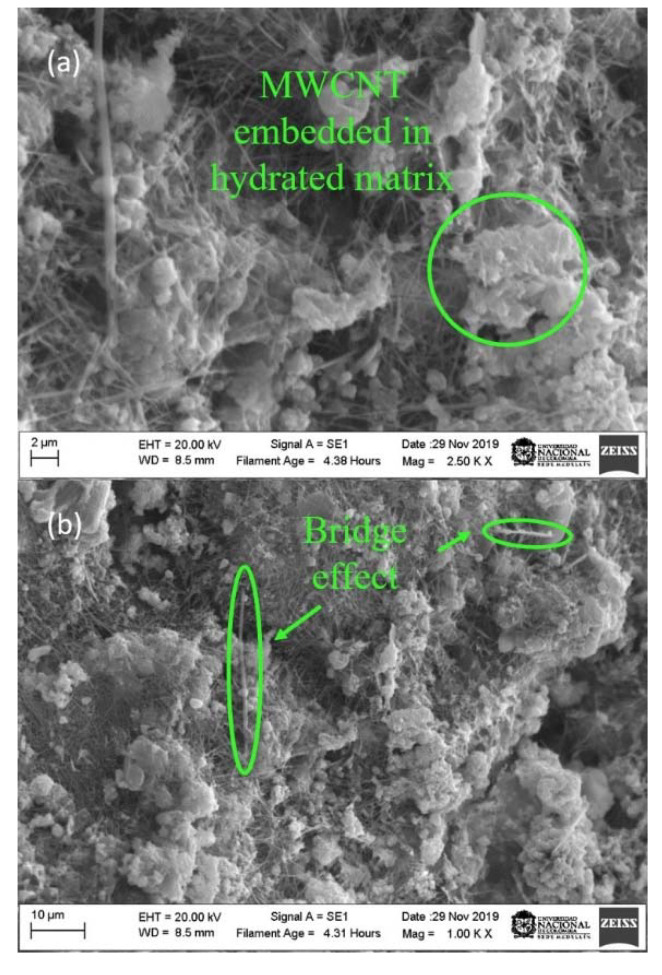
SEM images of Portland cement specimens with a dispersion solution addition at 490 J/g. (**a**) 2 µm (**b**) 10µm.

**Table 1 materials-13-04149-t001:** Time used for each energy of dispersion.

Energy (J/g)	90	190	290	340	390	440	490	590
Time (s)	552	1170	1744	2044	2346	2646	2947	3549

**Table 2 materials-13-04149-t002:** Summary of results from quasi-static compression tests on Portland cement specimens performed with an MWCNTs solution at one repose week.

	Cement Paste Cylinder	Cement Paste + H_2_O + TritonTX-100	Cement Paste with a Solution of H_2_O + TX-100 + MWCNTs Submitted to 190 J/g	Cement Paste with a Solution of H_2_O + TX-100 + MWCNTs Submitted to 390 J/g	Cement Paste with a Solution of H_2_O + TX-100 + MWCNTs Submitted to 490 J/g
Curing Time					
7 days	Elasticity Modulus (GPa)	2.36 ± 0.46	2.66 ± 0.11	2.96 ± 0.10	3.51 ± 0.25
	Maximum Strength (MPa)	15.44 ± 0.79	20.49 ± 0.21	22.29 ± 0.17	24.59 ± 0.24
14 days	Elasticity Modulus (GPa)	2.70 ± 0.85	2.75 ± 0.13	3.66 ± 0.27	3.73 ± 0.12
	Maximum Strength (MPa)	16.54 ± 0.68	27.84 ± 0.62	27.10 ± 0.33	28.58 ± 0.37
28 days	Elasticity Modulus (GPa)	2.84 ± 0.97	2.88 ± 0.04	3.89 ± 0.45	5.21 ± 0.11
	Maximum Strength (MPa)	19.54 ± 0.59	32.47 ± 0.45	33.18 ± 0.76	37.85 ± 0.99

**Table 3 materials-13-04149-t003:** Summary of results from quasi-static compression tests Portland cement specimens performed with an MWCNTs at four repose weeks.

	Cylinder Pastes	Cement Paste + H_2_O + TritonTX-100	Cement Paste with a Solution of H_2_O + TX-100 + MWCNTs Submitted to 190 J/g	Cement Paste with a Solution of H_2_O + TX-100 + MWCNTs Submitted to 390 J/g	Cement Paste with a Solution of H_2_O + TX-100 + MWCNTs Submitted to 490 J/g
Curing Time					
7 days	Elasticity Modulus (GPa)	2.36 ± 0.46	2.55 ± 0.26	3.02 ± 0.32	3.52 ± 0.07
	Maximum Strength (MPa)	15.44 ± 0.79	21.00± 0.25	22.28± 0.57	24.84 ± 0.96
14 days	Elasticity Modulus (GPa)	2.70 ± 0.85	2.85 ± 0.19	3.48 ± 0.29	3.64 ± 0.28
	Maximum Strength (MPa)	16.54 ± 0.68	22.77 ± 0.56	24.82 ± 0.25	28.21 ± 0.72
28 days	Elasticity Modulus (GPa)	2.84 ± 0.97	2.93 ± 0.02	3.87 ± 0.13	5.14 ± 0.31
	Maximum Strength (MPa)	19.54 ± 0.59	24.12 ± 0.60	28.46 ± 0.37	39.53 ± 0.64

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
