# Peer review of "Time-Stability Dispersion of MWCNTs for the Improvement of Mechanical Properties of Portland Cement Specimens"

_materials, 2020, doi:10.3390/ma13184149_

Round 1

Reviewer 1 Report

The papers about the possibility of new cementitious binders with  Multi-Wall Carbon Nanotubes. The acronym can be used in the text, but authors should avoid the use in the title. In general the English should be revised, the introduction needs some

Please revise the English, some sentences seem to be incomplete: eg lines 31,32.

Line 46: this is a crucial point, please insert some other bibliographic references as example Belli, A., Mobili, A., Bellezze, T., Tittarelli, F., & Cachim, P. (2018). Evaluating the self-sensing ability of cement mortars manufactured with graphene nanoplatelets, virgin or recycled carbon fibers through piezoresistivity tests. Sustainability, 10(11), 4013.

Also, the mechanical properties are discussed, so some researches about it should be cited.

Material and methods:

The mix design should be insert as a percentage of each material on the total weight: the total content of MWCNT should be explicit.

Line 55: formatting error: 3 is a superscript

Lines 63-64: authors should specify why 13 weeks are chosen as time.

Line 67:why only that samples are discussed? The reason should be explicit in the method section.

Which type of cement is used?

Line 152: why now 4 weeks of storage are considered? (also line 191) probably the authors would consider the curing of the specimens

Line159: when the changing in temperature is occurred in this experiment?

Line 169: how the modulus of elasticity is evaluated? Please add it in the methods section

Line 170: which type of mechanical resistance (and line 305)

Line 189: there are percentages and 3 values, the section is not clear, it should be rewritten

Section 3.2: Figures and tables should be placed in the better place (when cited).

Lines 209, 217: the acronym is not correct, c-s-h (as defined in line 32) should be added.

Line 290 authors  should type correctly axis ‘x’

The specific contribution of each author is missing

Author Response

The papers about the possibility of new cementitious binders with Multi-Wall Carbon Nanotubes. The acronym can be used in the text, but authors should avoid the use in the title. In general the English should be revised,

The Introduction

Please revise the English, some sentences seem to be incomplete: eg lines 31,32.

Response: Thank you for yor correction. The English was revised and corrected

Line 46: this is a crucial point, please insert some other bibliographic references as example Belli, A., Mobili, A., Bellezze, T., Tittarelli, F., & Cachim, P. (2018). Evaluating the self-sensing ability of cement mortars manufactured with graphene nanoplatelets, virgin or recycled carbon fibers through piezoresistivity tests. Sustainability, 10(11), 4013.

Response: the references were included with the phrase " The combination of these methods has been widely used in recent years, obtaining more productive and stable dispersions [24][25][26]."

[25] A. Belli, A. Mobili, T. Bellezze, F. Tittarelli, and P. Cachim, "Evaluating the self-sensing ability of cement mortars manufactured with graphene nanoplatelets, virgin or recycled carbon fibers through piezoresistivity tests," Sustain., vol. 10, no. 11, 2018.

[26] W. Yu and H. Xie, "A Review on Nanofluids : Preparation , Stability Mechanisms , and Applications," J. Nanomater., vol. 2012, pp. 1–17, 2012.

Also, the mechanical properties are discussed, so some researches about it should be cited.

Response: the references were included, and it was added the next sentence in the introduction section "… demonstrated a significant improvement of the mechanical resistance in the cementitious matrix, due to the filling of microcracks, pores and bonds between the surface and the matrix [8][9][10][11][12]"

[9] S. Lu et al., "The mechanical properties, microstructures and mechanism of carbon nanotube-reinforced oil well cement-based nanocomposites," RSC Adv., vol. 9, no. 46, pp. 26691–26702, 2019.
[10] C. G. N. Marcondes, M. H. F. Medeiros, J. M. Filho, and P. Helene, "Carbon Nanotubes in Portland cement concrete: Influence of dispersion on mechanical properties and water absorption," Alconpat, vol. 5, pp. 97–113, 2015.

Material and methods:

The mix design should be inserted as a percentage of each material on the total weight: the total content of MWCNT should be explicit.

Line 55: formatting error: 3 is a superscript

Response: the superscript was corrected

Lines 63-64: authors should specify why 13 weeks are chosen as time.

Response: The phrase "in a dark place for 13 weeks." was changed by "taking spectra in a UV-Vis spectrophotometer and Zetasizer Nano equipment, during 1, 2, 4, 10 and 13 weeks. The experiment was finished the thirteen weeks, because the MWCNTs were falling out abruptly in the solution."

Line 67: why only those samples are discussed? The reason should be explicit in the method section.

Response: in the materials and methods section is introduced the paragraph "Besides, using the RAMAN analysis, we obtained three characteristic zones, which are represented by 190 J/g, 390 J/g and 490 J/g dispersion energies and these energies were used to make the cement paste cylinders."

Which type of cement is used?

Response: The phrase " Cement paste cylinders were built with water-to-cement (w/c) ratio of 0.4 [33]. The MWCNTs and surfactant are part of the water ratio. The Cement used was Portland Cement type 1" was included in the materials and methods section.

Line 152: why now 4 weeks of storage are considered? (also line 191) probably the authors would consider the curing of the specimens

Response: In the materials and methods section, it was introduced the phrase "…, we study the stabilization over time, taking spectra in a UV-Vis spectrophotometer and Zetasizer Nano equipment, during 1, 2, 4, 10 and 13 weeks. The experiment was finished the thirteen weeks, because the MWCNTs were falling out abruptly in the solution." When we express four weeks of storage, we reference to the fourth week.

Line159: when the changing in temperature is occurred in this experiment?

Response: The phrase " Other aspect that could have influenced the irregular behavior of samples is the variation of temperature (observed for cycles on/off in the dispersion process), which affects the zeta potential [48]." was added.

Line 169: how the modulus of elasticity is evaluated? Please add it in the methods section

Response: In the Characterization methods section the following comment was introduced " The elasticity modulus and maximum strength were obtained from the stress-strain curve."

Line 170: which type of mechanical resistance (and line 305)

Response: the phrase "mechanical properties of compression" was included in different places of the document.

Line 189: there are percentages and 3 values, the section is not clear, it should be rewritten

Response: The section was re-written as " Table 3 presents the effects on the elasticity modulus from the addition of MWCNTs in the test samples, at different dispersion energies (190J/g, 390 J/g and 490 J/g) and a time of rest of the four weeks. Differences between the elasticity modulus of the specimens with and without MWCNTs are 5.61 ±2.44 % (190 J/g), 32.12 ±4.15 % (390 J/g) and 65.07 ±15.92 % (490 J/g). Meanwhile, for the maximum strength, changes are 29.72 ±6.28 (190 J/g), 44.97 ± 0.67 (390 J/g) and 81.59 ±20.71 (490 J/g)."

Section 3.2: Figures and tables should be placed in a better place (when cited).

Response: tables and figures were better located and cited

Lines 209, 217: the acronym is not correct, c-s-h (as defined in line 32) should be added.

Response: thanks a lot for your appreciation; it was a mistake and it was corrected.

Line 290 authors should type correctly axis 'x'

Response: the axis "x" was numbered

The specific contribution of each author is missing

Response: Jorge H Quintero-Orozco provided the main idea and composites development with applications into civil engineering

Laura Echeverry provided the development of the Methodology, making of dispersion to different energies

Preparation specimens of cement paste Natalia Alzate and Laura Echeverry

Characterization of the samples Laura Echeverry and Jorge H Quintero-Orozco

Analysis formal and results, each author.

Achievement of budget, Elisabeth Restrepo Parra, Rogelio Ospina and Jorge H Quintero-Orozco

Drafting and preparation of the document, Natalia Alzate, Laura Echeverry, Elisabeth Restrepo and Jorge H Quintero-Orozco.

Supervision of the document, Elisabeth Restrepo, Rogelio Ospina and Jorge H Quintero-Orozco

Developments of conclusions: Laura Echeverry, Natalia Alzate, Elisabeth Restrepo and Jorge H Quintero-Orozco.

Reviewer 2 Report

The authors studied the sonication process of CNT dispersion and investigated the stability and the correspondingly cement composites. Some suggestions are as follow:

(1) For a better background introduction of CNT and CNT dispersion process and CNT composites, following references are recommended:

https://doi.org/10.1002/adma.201902028

https://doi.org/10.1002/aelm.201800811

(2) The writing style of this paper may not be appropriate for the scientific presenting. Most of sentences are too long with sophisticated grammatical constructions that are not useful for the scientific idea convey. From the viewpoint of reviewer, authors are strongly suggested to reconstruct these sentences, some examples are as follow:

On page 3 line 112: In figure 3, the IG/ID ratio shows a decreasing trend to the average at the same time that the sonication energy increases due to the decrease in the intensity of the peak associated with the ordered C-C bonds, which causes an inversely proportional effect with the IG/ID ratio.

On page 4 line 158: Another aspect that could have influenced irregular behaviour is the variation of temperature, which ranged from 30°C to 66°C and the change in the pH, parameters that influence zeta potential

(3) The concentration of CNTs is an important factor that dictates the stability of the CNT suspension. Also as an additive, the concentration of CNTs in the final composite is also critical in terms of property enhancements. Therefore, considering these two aspects, how about the stability of the CNT suspension with a higher concentration? As using a higher concentration of CNT suspension is expected to achieve a better performance of composites.

(4) The amount of the T-X 100 surfactant is another key to obtain stable CNT suspension. More T-X 100 is expected to have a better dispersion capability for forming a stable CNT suspension. But too much T-X 100 is detrimental for the following process such as composite preparation. So how to balance this trade-off? Authors are suggested to add discussion on this point.

Author Response

The authors studied the sonication process of CNT dispersion and investigated the stability and the corresponding cement composites. Some suggestions are as follow:

For a better background introduction of CNT and CNT dispersion process and CNT composites, the following references are recommended:

https://doi.org/10.1002/adma.201902028

https://doi.org/10.1002/aelm.201800811

Response: the references were included in introduction section with the phrase " Meanwhile, the Multi-Wall Carbon Nanotubes (MWCNTs) have demonstrated a significant improvement of the mechanical resistance in the cementitious matrix, due to the filling of microcracks, pores and bonds between the surface and the matrix [8][9][10][11][12].

[8] S. Zhang et al., "Carbon-Nanotube-Based Electrical Conductors: Fabrication, Optimization, and Applications," Adv. Electron. Mater., vol. 5, no. 6, pp. 1–36, 2019.

[11] X. Zhang, W. Lu, G. Zhou, and Q. Li, "Understanding the Mechanical and Conductive Properties of Carbon Nanotube Fibers for Smart Electronics," Adv. Mater., vol. 32, no. 5, pp. 1–21, 2019.

(2) The writing style of this paper may not be appropriate for the scientific presenting. Most of sentences are too long with sophisticated grammatical constructions that are not useful for the scientific idea convey. From the viewpoint of reviewer, authors are strongly suggested to reconstruct these sentences, some examples are as follow:

On page 3 line 112: In figure 3, the IG/ID ratio shows a decreasing trend to the average at the same time that the sonication energy increases due to the decrease in the intensity of the peak associated with the ordered C-C bonds, which causes an inversely proportional effect with the IG/ID ratio.

Response: The paragraph was re-written by " With the IG/ID ratio, a relationship of the induced disorder with the increase in sonication energy was found. In figure 3, the IG/ID ratio shows a decreasing trend when the sonication energy increases, and it is associated with the ordered C-C bonds, which causes an inversely proportional effect with the IG/ID ratio. Three zones are observed: i) the first zone between 90 and 290 J/g, shows a significant decrease in the IG/ID ratio from 1 to 0.75 [34]; this can be associated to the sonication process, which causes alterations in the structure of MWCNTs, because are unstable molecules (as can be seen in figure 4); ii) In the second zone, from 290 to 440 J/g, the induced energy is greater than the energy between bonds π, which causes that the bonds break, without reaching the maximum energy allowed and iii) in the third zone between 440 and 590 J/g, the IG/ID ratio decreases again, inducing a better dispersion. However, in the literature, these dispersions cannot be considered as the most optimal energies, because that can contribute to the breaking and fragmentation of MWCNTs. The induced energy affects the π and the sigma bonds of the structure. It implying loss of characteristic bonds that provide the unique properties of nanotubes [41].

On page 4 line 158: Another aspect that could have influenced irregular behaviour is the variation of temperature, which ranged from 30°C to 66°C and the change in the pH, parameters that influence zeta potential

Response: the paragraph was re-written as follows " Figure 5 shows the changes of the absorbance spectrum taken the first week. The characteristic peak of the individual MWCNTs is observed at a wavelength of approximately 300 nm [45]. At sonication energies below 190 J/g, the peak intensity remains constant, while at energies above this value, the peak exhibits a significant increase. That phenomenon can be due to two simultaneous effects in the sonication process: i) the deagglomeration of the MWCNTs and ii) fragmentation of the individual MWCNTs [22]”.

Besides the paper was rewritten carefully

(3) The concentration of CNTs is an essential factor that dictates the stability of the CNT suspension. Also, as an additive, the concentration of CNTs in the final composite is critical in terms of property enhancements. Therefore, considering these two aspects, how about the stability of the CNT suspension with a higher concentration? As using a higher concentration of CNT suspension is expected to achieve a better performance of composites.

Response: This work is a derivate of a master thesis. the study of dispersants concentrations was carried out; nevertheless, we decided to present those results in other work, since on the contrary, the paper woulb be very long. Your suggestions are very valuated for us because the better dispersion was to 10 mM of molarity (it is used in this paper).

Now, the CNTs concentrations was based in work “Influence of MWCNT/surfactant dispersions on the rheology of Portland cement pastes. Oscar A. Mendoza Reales, Yhan Paul Arias Jaramillo, Juan Carlos Ochoa Botero, Carmen Alicia Delgado, Jorge Hernán Quintero, Romildo Dias Toledo Filho. Cement and Concrete Research Volume 107, May 2018, Pages 101-109. https://doi.org/10.1016/j.cemconres.2018.02.020.

But we think that it is necessary to make a work about the different concentration of CNT, thanks a lot for your appreciations.

(4) The amount of the T-X 100 surfactant is another key to obtain stable CNT suspension. More T-X 100 is expected to have a better dispersion capability for forming a stable CNT suspension. But too much T-X 100 is detrimental for the following process such as composite preparation. So how to balance this trade-off? Authors are suggested to add discussion on this point.

Response: The response is in the previous question

Reviewer 3 Report

Time-stability dispersion of MWCNT for the improvement of mechanical properties of Portland cement specimens

Laura Echeverry, Natalia Alzate, Elisabeth Restrepo-Parra, Rogelio Ospina, Jorge Quintero

The paper is about the evaluation of the stability in time of dispersion in water suspensions with MWCNTs.  The topic is interesting for possible applications in cement-based nanocomposites. However, ony a dispersant is considered, and it is not clear the real importance of long-term stability of MWCNTs water suspensions because when the cement is added, and the solidification occurs, which is the importance of the stability in water?

Also, why don’t investigating different amount of surfactant. Could it affect the results?

The reviewer suggests to add more references, above all about the dispersion approaches an the types of dispersants present in literature, in order to clarify the original contribution of the paper and its real novelty and importance with respect to possible applications.

The reviewer recommends to check the paper, modify the phrases that are not clear, control subscript and superscript, and strongly suggests to improve English. Also, please check the typos and punctuation, with particular attention to uppercases.

The reviewer suggests to describe better the figures.

The reviewer strongly suggests to prove and explain all the strange results in the paper and the decisions taken to consider different variables in different tests (different energies, timing, …), as indicated in the attached revised manuscript.

Further comments are also recommended, above all about the original contribution and the applications of the results of the research.

The reviewer strongly recommends to add comments in the sections of the discussion of the results and of conclusions, in order to identify the best mix, its possible applications and the impact that such a result could have in real situations. Indeed, it is not clear the real impact of this research, which results not so significant in the scientific panorama.

The authors, investigate only one type of MWCNTs and one type of surfactant: could the results be generalized? And how? This discussion could make the paper more significant.

Further recommendations are reported in the attached file.

Author Response

The paper is about the evaluation of the stability in time of dispersion in water suspensions with MWCNTs.  The topic is interesting for possible applications in cement-based nanocomposites. However, only a dispersant is considered, and it is not clear the real importance of long-term stability of MWCNTs water suspensions because when the cement is added, and the solidification occurs, which is the importance of the stability in water?

Response: In industrial applications, we have noted that to make the sonication process before being adhered to cement paste is not rentable, because the schedule of the process is increasing. It is necessary to develop significant amounts of dispersion, and this dispersion needs long time to preparation. Besides, if any process is not possibly to made, the sonication is lost, because it is not known the time of stabilization.

Also, why don't investigating different amount of surfactant. Could it affect the results?

Response: In previous works, we study three types of surfactants (sodium dodecyl sulfate, cetylpyridinium chloride and triton TX-100) to prepare cement pastes with and without MWCNTs.

  1. “Influence of MWCNT/surfactant dispersions on the rheology of Portland cement pastes. Oscar A. Mendoza Reales, Yhan Paul Arias Jaramillo, Juan Carlos Ochoa Botero, Carmen Alicia Delgado, Jorge Hernán Quintero, Romildo Dias Toledo Filho. Cement and Concrete Research Volume 107, May 2018, Pages 101-109. https://doi.org/10.1016/j.cemconres.2018.02.020.
  2. Reinforcing Effect of Carbon Nanotubes/Surfactant Dispersions in Portland Cement Pastes. Oscar A. Mendoza Reales, Caterin Ocampo, Yhan Paul Arias Jaramillo, Juan Carlos Ochoa Botero, Jorge Hernán Quintero, Emílio C. C. M. Silva, and Romildo Dias Toledo Filho. Advances in Civil Engineering Volume 2018, Article ID 2057940. https://doi.org/10.1155/2018/2057940.
  3. Influence of MWCNT/surfactant dispersions on the mechanical properties of Portland cement pastes. B Rodríguez, J H Quintero, Y P Arias, O A Mendoza-Reales, J C Ochoa-Botero and R D Toledo-Filho. Journal of Physics: Conference Series, Volume 935, 012014. https://doi.org/10.1088/1742-6596/935/1/012014

The reviewer suggests to add more references, above all about the dispersion approaches an the types of dispersants present in literature, in order to clarify the original contribution of the paper and its real novelty and importance with respect to possible applications.

Response: the next references were included in the introduction section: 

[12] K. M. Liew, M. F. Kai, and L. W. Zhang, "Carbon nanotube reinforced cementitious composites: An overview," Compos. Part A Appl. Sci. Manuf., vol. 91, no. October, pp. 301–323, 2016

[16] E. Batiston, “Estudo exploratório dos efeitos de nanotubos de carbono em matrizes de cimento Portland,” Universidad Federal de Santa Catarina, 2007.

[20] S. J. Chen, C. Y. Qiu, A. H. Korayem, M. R. Barati, and W. H. Duan, "Agglomeration process of surfactant-dispersed carbon nanotubes in unstable dispersion: A two-stage agglomeration model and experimental evidence," Powder Technol., vol. 301, pp. 412–420, 2016.

The reviewer recommends checking the paper, modify the phrases that are not clear, control subscript and superscript, and strongly suggests to improve English. Also, please check the typos and punctuation, with particular attention to uppercases.

Response: The appreciations of the pdf document were made carefully.

The reviewer suggests describing better the figures.

Response: the figures were revised and improved.

The reviewer strongly suggests proving and explain all the strange results in the paper and the decisions taken to consider different variables in different tests (different energies, timing, …), as indicated in the attached revised manuscript.

Response: The appreciations of the pdf document were made carefully.

Further comments are also recommended, above all about the original contribution and the applications of the results of the research.

Response: The conclusions section was re-written and it was adhered the next sentence "Finally, for applications in civil engineering, it was found that the solution composed by H2O+TX-100-MWCNTs shows a suitable stability, when it was stored during four weeks, making this kind of researches of potential interest for this industry"

The reviewer strongly recommends to add comments in the sections of the discussion of the results and of conclusions, in order to identify the best mix, its possible applications and the impact that such a result could have in real situations. Indeed, it is not clear the real impact of this research, which results not so significant in the scientific panorama

Response: the next sentence was added in the abstract: "Finally, we found an improvement of the mechanical properties of the samples built with Portland cement and solutions stored for one and four weeks; it can be concluded that the MWCNTs improved the hydration period"

The conclusions section was re-written and it was added the next sentence "Finally, for applications in civil engineering was found a solution composed by H2O+TX-100-MWCNT that can be stored by four weeks, which make that the research in this area can be potentialized to industrial escalations."

The authors, investigate only one type of MWCNTs and one type of surfactant: could the results be generalized? And how? This discussion could make the paper more significant.

Response: With the results here presented, we could not generalize these results to different MWCNTs and Surfactants. For example, if it is chosen other non ionic surfactant, the electrostatic charges influence in agglomerations.

Further recommendations are reported in the attached file. peer-review-8152760.v1.pdf

Reviewer 4 Report

The study focuses on the dispersion evaluation of CNTs in water, in order to be used in cement applications.

The design of experiments is sound and appropriate. However, the Introduction should be elaborated and further studies that are related with the incorporation of CNTs in cement to be included (taking into account the dispersion methodology and the results in mechanical properties).

The following minor issues need to be taken into account:

  • The fragmentation of the CNTs after ultrasonication it would be better to be examined with TEM and not only with RAMAN measurements.
  • The DLS measurement has been proved to be not appropriate for measuring high aspect ratio materials, such as CNTs. The use of the method should be justified.
  • The use of aqua Milli q water is not representative for the application in real conditions.

The comments below refer to the structure of the manuscript and editing issues:

Paragraph 3.1. analysis and Discussion: Please use capital "a"

Paragraph 3.2. Please ensure according to the guide of authors that figures and tables are placed at the end of the manuscript, otherwise they should be placed next to the text they are referred to.

References: Editing is needed since numbers in brackets have remained in several lines.

Author Response

The study focuses on the dispersion evaluation of CNTs in water, in order to be used in cement applications.

The design of experiments is sound and appropriate. However, the Introduction should be elaborated and further studies that are related with the incorporation of CNTs in cement to be included (taking into account the dispersion methodology and the results in mechanical properties).

Response: in the Introduction the next paragraph was added " Firstly, we made a dispersion of MWCNTs in water MilliQ (ultrapure water of laboratory) and Triton TX-100 surfactant through an ultrasonic tip, and after that, we study the stabilization over time, taking spectra in a UV-Vis spectrophotometer and Zetasizer Nano equipment, during 1, 2, 4, 10 and 13 weeks. The experiment was finished the thirteen weeks, because the MWCNTs were falling out abruptly in the solution. To evaluate the structural damage induced in the first week, RAMAN microscopy was used. Later, it was prepared two series of cement paste cylinders, at first and fourth weeks after the sonication process, observing that the mechanical properties are maintained. It was considered the fourth week since, for later weeks, the dispersion fall-out abruptly, and the nanotubes were agglomerated. Finally, it was obtained elastic modulus of the cylinders through a Humbolt HM 5030 Master Loader equipment. Experimental parameters of the dispersion were based on some previous works carried out by us"

The following minor issues need to be taken into account:

The fragmentation of the CNTs after ultrasonication it would be better to be examined with TEM and not only with RAMAN measurements.

Response: We are entirely according with the referee; nevertheless, at this moment, we do not have the possibility of having a TEM equipment, only we have the RAMAN equipment. Then, we used all the techniques that we have available. Withn this techiniques, we can obtain morphology, composition and mechanical porperties, that allow us to have a wide inside of the samples behaviour

The DLS measurement has been proved to be not appropriate for measuring high aspect ratio materials, such as CNTs. The use of the method should be justified.

Response: The equipment used to make the electron affinity measurements is a Zetasizer, which has different measurements method. We used the Zeta Potential method, and it does not require the DLS mode.

The use of aqua Milli q water is not representative for the application in real conditions.

Response: Completely according with you, but due to than the Triton TX-100 is a surfactant no ionic, we chose the MilliQ water for it did not problems of electronegativities in the solution. For next researches, we are going to consider it.

The comments below refer to the structure of the manuscript and editing issues:

Paragraph 3.1. analysis and Discussion: Please use capital "a"

Response: the correction was applied

Paragraph 3.2. Please ensure according to the guide of authors that figures and tables are placed at the end of the manuscript, otherwise they should be placed next to the text they are referred to.

Response: tables and figures were better located and cited.

References: Editing is needed since numbers in brackets have remained in several lines.

Response: It was made the edition carefully in the document.

Round 2

Reviewer 3 Report

Time-stability dispersion of MWCNT for the improvement of mechanical properties of Portland cement specimens

Laura Echeverry, Natalia Alzate, Elisabeth Restrepo-Parra, Rogelio Ospina, Jorge Quintero

The authors addressed most of the suggestions indicated by the reviewer.

The paper has been improved.

The reviewer recommends to current the typos indicated in the attached file.

Author Response

  • “about of the properly disperse the MWCNTs” please modify

Reply: It was modified as  “how to properly disperse MWCNTs”

  • “The cylinders were builtara el under the NTC 550” please correct

Reply: It was modified as  “The cylinders were built following the NTC 550”

  • Van der Walls

Reply: The name was already adapted in the modifications

  • [15] A. D. Filippo Ubertini, Simon Laflamme, Innovative Developments of Advanced Multifunctional Nanocomposites in Civil and Structural Engineering, no. February 2016. 2016. Please use only the first letter for the name and check all the authors

Reply: Citation corrected in the manuscript

[15] K. Loh, S. Nagarajaiah, Innovative Developments of Advanced Multifunctional Nanocomposites in Civil and Structural Engineering, Woodhead Publishing, Cambridge, 2016.

  • [27] Oscar A Mendoza Reales, Yhan Paul Arias Jaramillo, Juan Carlos Ochoa Botero, Carmen Alicia Delgado, Jorge Hernán Quintero, Romildo Dias Toledo Filho, “ Influence of MWCNT/surfactant dispersions on the rheology of Portland cement pastes”, Cem. Concr. Res., 107, no. February, pp. 101–109, 2018. please use only the first letter for the names and check.

Reply: Citation corrected in the manuscript

[27] O. A. M. Reales, Y. P. A. Jaramillo, J. C. O. Botero et al., “Influence of MWCNT/surfactant dispersions on the rheology of Portland cement pastes,” Cem. Concr. Res., vol. 107, pp. 101–109, 2018